# Point Cloud Upsampling With Geometric Algebra Driven Inverse Heat Dissipation

## ABSTRACT

Point cloud upsampling is crucial for 3D reconstruction, with recent research significantly benefitting from the advances in deep learning technologies. The majority of existing methods, which focus on a sequence of processes including feature extraction, augmentation, and the reconstruction of coordinates, encounter significant challenges in interpreting the geometric attributes they uncover, particularly with respect to the intricacies of transitioning feature dimensionality. In this paper, we delve deeper into modeling Partial Differential Equations (PDEs) specifically tailored for the inverse heat dissipation process in dense point clouds. Our goal is to detect gradients within the dense point cloud data distribution and refine the accuracy of interpolated points' positions along with their complex geometric nuances through a systematic iterative approximation method. Simultaneously, we adopt multivectors from geometric algebra as the primary tool for representing the geometric characteristics of point clouds, moving beyond the conventional vector space representations. The use of geometric products of multivectors enables us to capture the complex relationships between scalars, vectors, and their components more effectively. This methodology not only offers a robust framework for depicting the geometric features of point clouds but also enhances our modeling capabilities for inverse heat dissipation PDEs. Through both qualitative and quantitative assessments, we demonstrate that our results significantly outperform existing state-of-the-art techniques in terms of widely recognized point cloud evaluation metrics and 3D visual reconstruction fidelity.

## CCS CONCEPTS

• **Computing methodologies** → **Reconstruction**; *Shape representations*.

## KEYWORDS

Point cloud upsampling, Partial Differential Equations, Geometric algebra

## 1 INTRODUCTION

Point clouds serve as the foundational fabric for 3D modeling[9, 12, 21], rendering intricate details of real-world environments. While capturing these clouds is now more accessible thanks to advanced scanning technologies, the raw data often presents itself as sparse and inconsistent, with noise that undermines its utility in downstream applications such as autonomous navigation[13, 38] and virtual/augmented reality[7, 33]. Upsampling is thus not merely a refinement but a necessity to attain a dense, accurate representation of the spatial continuum.

Classical methods for point cloud upsampling primarily resort to optimization-based approaches [1, 10, 11, 16, 40]. These methods utilize shape priors, including global structures which are defined as objective energy functions to shape and refine the upsampling results. However, shape priors that assume accurate normal estimation or the presence of a smooth surface in the local geometry limit optimization-based methods in representing complex and massive point cloud data and result in degraded reconstruction performance.

Recently, deep learning-based techniques have been widely employed in point cloud upsampling [14, 15, 18, 26, 28, 42, 43]. Deep learning based point cloud sampling usually consists of three steps, including feature extraction that captures point-wise semantics from low-resolution point clouds, feature expansion tailored to a specific upsampling rate, and 3D coordinate prediction for upsampled points based on the expanded features. Convolutional neural networks were first adopted to extract global features from point clouds [25] but suffer from projection distortion due to the incapability to represent local geometry. Graph convolutions like EdgeConv [23] were then leveraged to ensure the invariance to point ordering but cannot accurately capture local geometric shapes such as surface distances with discretized graphs. Recently, point transformers [45] have been leveraged to achieve improved performance by exploiting the correlations between points using matrices of learnable parameters. However, similarity measurement through inner product between high-dimensional vector features neglects the geometry like angles and oriented planes of the elements in the high-dimensional spaces and leads to outliers or artifacts in the upsampled point clouds. Furthermore, deep learning based methods rely on the upsampling rate and require various models for different rates. These problems underscore the urgent need for deep learning models with enhanced generalization capabilities and error resilience in point cloud upsampling.

The emergence of generative diffusion models [8, 19, 36] suggests the potential of considering 2D/3D signal processing from the perspective of statistical modeling. Diffusion models learn to generate data by gradually transforming noise into structured output through a gradual process like the Markov chain. This can capture complex data distributions more finely than models that attempt to learn this in one single step and do not suffer from mode collapse as GAN [5]. Denoising Diffusion Probabilistic Models (DDPMs) [8] rely on the distribution of training data. In 2D image processing, where the data lies on a grid with fixed scale and have no specific geometric shapes, DDPMs are well-suited for inferring high-quality images entirely from the predicted distribution. However, point clouds represent a distribution with defined geometric shapes. If

Permission to make digital or hard copies of all or part of this work for personal or classroom use is granted without fee provided that copies are not made or distributed for profit or commercial advantage and that copies bear this notice and the full citation on the first page. Copyrights for components of this work owned by others than the author(s) must be honored. Abstracting with credit is permitted. To copy otherwise, or republish, to post on servers or to redistribute to lists, requires prior specific permission and/or a fee. Request permissions from permissions@acm.org.
*ACM MM, 2024, Melbourne, Australia*
© 2024 Copyright held by the owner/author(s). Publication rights licensed to ACM.
ACM ISBN 978-x-xxxx-xxxx-x/YY/MM
https://doi.org/10.1145/nnnnnnn.nnnnnnn

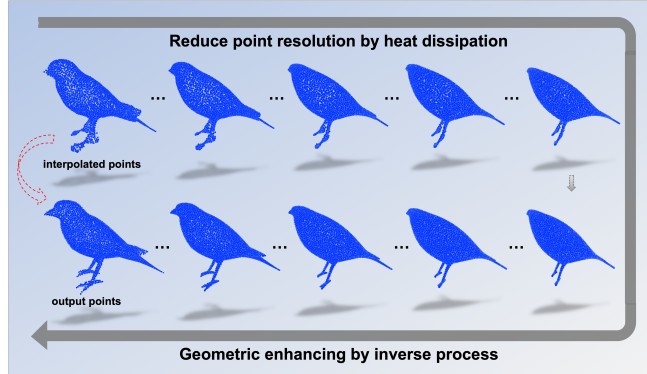

**Figure 1: Example of the heat dissipation forward process and the inverse process for point cloud.**

the training data does not adequately represent the geometric variations in the test point clouds, the model might struggle to accurately infer and reconstruct the details and structures within the point cloud. To address this issue, we have moved away from adding noise, a process that indiscriminately disrupts geometric configurations. Instead, we are considering a diffusion scheme that allows for the geometric shapes to be traceable and more predictably handled.

Inspired by the efficient generative process of diffusion models [30], in this paper, we design PDE-based heat dissipation as forward diffusion process distinct from the previous methods. The defining characteristic of our PDE-based forward process is its commitment to a regular, traceable degradation of the point cloud's geometric shapes. It meticulously maintains the dataset's essential geometrical features—such as curvature and continuity. This careful approach guarantees that the diffusion process adheres to the original structure of the dataset. The alignment with the point cloud's inherent design is demonstrated in Figure 1.

Simultaneously, to more precisely capture the geometric features of point clouds, we have broadened point feature space from the traditional vector fields used in linear algebra to the more complex multivector fields of geometric algebra. Unlike the inner product attributes of vector fields, multivector fields incorporate scalar, vector, and higher-order elements to provide crucial information about the magnitudes, angles, and oriented planes of the elements involved in geometric product multiplication. During the training of our networks, we focus on learning the gradients of geometric distribution changes under the dissipation of point cloud heat. Accordingly, each point corresponds to a non-linear trajectory that includes both direction and length. Representation via geometric algebra can accommodate to the complex, multi-dimensional geometric nature of the point data to achieve upsampling. This methodology not only enhances the ability to discern subtle geometric nuances but also significantly improves the model's overall capabilities in handling complex spatial data.

Similar to conditional generative models, our point cloud upsampling model employs upsampled point clouds, obtained through rough linear interpolation, as a condition while exploiting prior knowledge. These coarsely interpolated point clouds undergo a forward heat dissipation process, followed by several iterations of

approximation through a reverse heat dissipation network, ultimately producing high-quality, dense point clouds.

The contributions of this paper are summarized as below.

- We have developed a PDE-based heat dissipation forward process specifically for point clouds and have examined its corresponding inverse heat dissipation as a reverse process to refine geometric details in point cloud upsampling tasks.
- We represent the geometric features in the multivector fields via geometric algebra for enhanced ability to capture geometric information through the interactions of multivector components and their unique geometric products.
- Comprehensive experiments demonstrate the outstanding capability of our work in generating geometric details in public benchmarks of point cloud upsampling.

## 2 RELATED WORK

### 2.1 Optimization-based Point Upsampling

In early research, point cloud upsampling was approached as an optimization problem. Alexa *et al.* [1] generate new points by constructing a Voronoi diagram vertices in the local tangent space. Lipman *et al.* [16] crafted an innovative locally optimal projection operator, leveraging the $L_1$ median for robust point resampling and surface reconstruction, exhibiting resilience to noise and outliers. Subsequently, the weighted Locally Optimal Projection (LOP) method [10] implemented an iterative process for normal estimation to refine the consolidated upsampled points. Huang *et al.* [11] advanced the field by introducing a progressive technique termed EAR, designed for edge-aware resampling of point sets. Wu *et al.* [40] employed a joint optimization approach for simultaneously refining inner points and surface points within their innovative point set representation framework. However, the efficacy of these methods often hinges on strong a priori assumptions, such as reliable normal estimation or inherent smoothness in local geometry. Consequently, they may falter when faced with complex and voluminous point cloud data.

### 2.2 Learning-based Point Upsampling

The fusion of deep learning, boasting powerful data-driven and trainable attributes, has significantly propelled advancements in the field of 3D data processing. Leveraging the potent representational capacities of deep neural networks has rendered the direct feature learning from 3D data feasible, as evidenced by pioneering architectures like PointNet[24], PointNet++[24], and EdgeConv[23].

PU-Net [43][43] stands at the forefront of incorporating deep neural networks into the realm of point cloud upsampling. It innovatively aggregates multi-scale features for each point using multiple Multilayer Perceptrons (MLPs), subsequently employing a channel shuffle layer to expand these features into an upsampled point cloud set. MPU [42] introduces a feature extractor based on EdgeConv[23], and enhances feature representation by assigning unique 1D codes for expansion. PUGAN[14] leverages adversarial training and devises an up-down-up unit to refine expanded features. PUGeoNet [27] innovates by initially generating points in 2D space before mapping them to 3D space. Meanwhile, PU-GCN[26] introduces the Inception DenseGCN for nuanced feature

extraction and employs NodeShuffle for subsequent feature expansion. NePs [4] posits that sampling points from a 2D continuous space can yield results of higher quality. Additionally, Grad-PU [6] reconceptualizes point cloud upsampling as a task of coordinate approximation, thereby obviating the need for designing specific upsampling modules. these methods usually have two aforementioned issues: fixed upsampling rate after each training and outliers or shrinkage artifact due to the difficulty of 3D coordinate estimation. Despite a few recent methods break the former limitation by meta-learning[41], the latter problem still remains unsolved.

## 2.3 Diffusion Models for Point Processing

Motivated by the achievements in 2D tasks[31, 32], 3D point cloud processing has increasingly embraced the capabilities of Denoising Diffusion Probabilistic Models (DDPM). [19] marks a pioneering endeavor in leveraging Denoising Diffusion Probabilistic Models (DDPM) for unconditional point cloud generation. Subsequently, [46] expand the use of DDPM for point cloud completion tasks by utilizing a point-voxel CNN [17] during the training phase. However, the process of voxelization adds a layer of computational complexity. Furthermore, PDR [24] takes raw point clouds as input. But this requires training the two stages (coarse-to-fine) of diffusion models, resulting in a greater time overhead. Moreover, PDR [20] accepts raw point clouds as input, necessitating the training of two stages (coarse-to-fine) of diffusion models, which incurs a significant increase in time overhead. Recently, a conditional denoising diffusion probability model PUDM [29] for point cloud upsampling has been introduced, innovatively leveraging sparse point clouds as a condition to directly model the gradient of data distribution. This approach facilitates the learning of geometric shapes without necessitating an additional upsampling module.

While DDPM showcases some advantageous attributes within point processing, it also harbors certain potential limitations: Firstly, while DDPM's auto-regressive properties enable robust modeling for objects at fixed scales with no geometric , they falter in producing high-quality point samples at flexible scales during inference, and treating multi-scale point upsampling as separate tasks leads to prohibitive training expenses. Secondly, point cloud generation networks conditioned on DDPM suffer from inadequate geometric feature perception due to insufficient effective prior knowledge, compromising the quality of generated results, despite attempts at mitigation through cost-intensive two-stage training methods.

## 3 PROPOSED METHOD

In this section, we first establish the formulation for the PDE-based heat dissipation applicable to both the forward and reverse diffusion processes in point clouds. Subsequently, we articulate the representation of point clouds utilizing multivectors derived from geometric algebra. Lastly, we formulate the objective that guide the training of our model.

## 3.1 PDE-based heat dissipation Formulation

### 3.1.1 motivation.

The conventional approach of Denoising Diffusion Probabilistic Models[8] (DDPM) is such that, over time, points progressively disperse into a disordered assembly. This phenomenon is known as the diffusion process, which morphs the original distribution into one of noise. However, as described in Section 2.3, the noise addition approach is not the best choice for point cloud geometric reconstruction.

We focus reducing point resolution, a less-explored aspect where scaling typically relies on straightforward point sub-sampling. [2, 39] introduced an alternative approach by executing a Partial Differential Equation (PDEs) which characterizes heat dissipation. Similarly point subsampling, the heat equation smooth the point cloud and removes fine detail, but an arbitrary amount of effective resolutions is allowed without explicitly decreasing the number of points. Thus We investigate diffusion-type point upsampling models based on directly reversing the heat dissipation and thus increasing the effective point resolution. The intuition is that as the original geometric details is erased in the forward process, a corresponding stochastic reverse process produces plausible reconstructions, defining a reconstruction model. Samples from the prior distribution are easy obtain due to the low dimensionality of low-resolution point cloud, and we adopt a training data based kernel density estimate.

### 3.1.2 PDE-based heat dissipation Formulation.

The formulation of the forward process contract points into a lower-dimensional subspace. We define it with the heat equation, a partial differential equation (PDE) that describes the dissipation of heat:

$$\frac{\partial \mathbf{u}(c, t)}{\partial t} = \Delta \mathbf{u}(c, t), \qquad (1)$$

where $c : \mathbb{R}^3$ are coordinates and $u : \mathbb{R}^3 \times \mathbb{R}_+ \to \mathbb{R}$ represents the idealized distribution of a specific attribute (such as heat or intensity) in a continuous 3D space, and $\Delta$ denotes the Laplace operator. This process operates independently on each point within the point cloud, employing Neumann-like boundary $(\partial u / \partial x = \partial u / \partial y = \partial u / \partial z = 0)$ conditions to manage behaviors at the cloud boundaries. As time progresses, every point in the cloud naturally gravitates towards its low-resolution state, thus smoothing the cloud and removing fine details at a larger scale, yet permitting variations in effective resolution without a significant reduction in the number of points.

Although Rissanen et al. [30] previously proposed a 2D generative model based on heat dissipation and modeled the process as a partial differential equation (PDE), they utilized a time-independent eigenbasis of the Laplace operator to solve the equation. This solution approach crudely smooths the 2D image to a uniform average. Consequently, we can interpret their process as completely deterministic degradations. In addressing the unique geometric characteristics of point clouds, We employ the Implicit Euler Method, integrating it seamlessly with standard PDE solvers for effective resolution of the equation. In this context, $\tau$ represents a small time increment, and $I$ is the identity matrix, establishing a stable and accurate computational step from $t$ to $t + \tau$ through the equation:

$$(\mathbf{I} - \tau \Delta \mathbf{u}(c, t)) \mathbf{u}(c, t + \tau) = \mathbf{u}(c, t) \qquad (2)$$

Consequently, the PDE model expressed in Equation (2) can be reformulated into an evolutionary equation:

$$\mathbf{u}(c, t) = \mathbf{u}(c, 0) + \int_0^t \Delta \mathbf{u}(c, \tau) d\tau \qquad (3)$$

For the sake of simplification, Equation (3) can be represented as $\mathbf{u}(c, t) = \mathcal{H}(t)\mathbf{u}_0(c)$, where $\mathcal{H}$ encapsulates a nonlinear evolutionary operator.

Thus we define the time steps $t_1, t_2, \ldots, t_K$ that correspond to latent variables $u(k)$, each of which has the same dimensionality as the data $u(0)$. Our forward process, or formally the variational approximation in the latent variable model, is defined as:

$$q\left(\mathbf{u}_{1:K} \mid \mathbf{u}_0\right) = \prod_{k=1}^{K} q\left(\mathbf{u}_k \mid \mathbf{u}_0\right) = \prod_{k=1}^{K} \mathcal{N}\left(\mathbf{u}_k \mid \mathcal{H}\left(t_k\right)\mathbf{u}_0, \sigma^2\epsilon\right) \quad (4)$$

Following the approach of Rissanen *et al.* [30], we disrupt the reversibility by infusing a modest quantity of noise, characterized by a standard deviation $\sigma$, into the forward process. This integration acknowledges the existence of branching into multiple plausible reverse pathways which is proved reasonable in [35].

We have access to a dataset comprising sparse-dense point pairs, $P = \{(u_s, x_d)\}$, each initiating from an unspecified conditional distribution $p(z|u_s)$. Rather than employing a network to independently train for the point cloud feature $z$, we adopt Midpoint Interpolation, as per He et al. [6], to coarsely upsample the point clouds. This technique also accomplishes the steps to achieve various upsampling rates, thereby obviating the need for subsequent network processes to consider elevation in point cloud dimensions. This marks a significant departure from the attribute of classical generative models which feature a one-to-many mapping, wherein a multitude of target images may align with a single source image. Our interest lies in learning a parametric approximation to $p(x_d|u_s, z)$ via an iterative refinement process that maps a source point set $u_s$ to a target point set $x_d$.

### 3.1.3 Training Objective.

The reverse, or geometric reconstruction process, is formulated as a Markov chain that starts with the prior state $\mathbf{u}_K$ and ends at the observed variable $\mathbf{u}_0$. We define it with conditional distributions:

$$p_\theta\left(\mathbf{u}_{0:K} \mid z\right) = p\left(\mathbf{u}_K \mid z\right) \prod_{k=1}^{K} p_\theta\left(\mathbf{u}_{k-1} \mid \mathbf{u}_k, z\right)$$

$$= p\left(\mathbf{u}_K \mid z\right) \prod_{k=1}^{K} \mathcal{N}\left(\mathbf{u}_{k-1} \mid \mu_\theta\left(\mathbf{u}_k, k, z\right), \delta^2\epsilon\right) \quad (5)$$

Here, $\theta$ represents the model's parameters, and $\delta$ signifies the standard deviation of the noise introduced at each step of the reverse process. Our objective is to optimize the marginal likelihood of the data $p(\mathbf{u}_0)$, which is expressed as $p(\mathbf{u}_0) = \int p_\theta(\mathbf{u}_0|\mathbf{u}_{1:K}, z) p_\theta(\mathbf{u}_{1:K}|z) d\mathbf{u}_{1:K}$. By adopting a Variational Autoencoder (VAE)-style evidence lower bound (ELBO) on this marginal likelihood and specifying the generative and inference distributions, we derive the following expression:

$$-\log p_\theta\left(\mathbf{u}_0|z\right) \leq \mathbb{E}_q\left[-\log \frac{p_\theta(\mathbf{u}_{0:K}|z)}{q(\mathbf{u}_{1:K}|\mathbf{u}_0)}\right]$$

$$= \mathbb{E}_q\left[-\log \frac{p_\theta(\mathbf{u}_K|z)}{q(\mathbf{u}_K|\mathbf{u}_0)} - \sum_{k=2}^{K} \log \frac{p_\theta(\mathbf{u}_{k-1}|\mathbf{u}_k, z)}{q(\mathbf{u}_{k-1}|\mathbf{u}_0)} - \log p_\theta\left(\mathbf{u}_0 \mid \mathbf{u}_1, z\right)\right]$$

$$= \mathbb{E}_q\left[L_K + \sum_{k=2}^{K} L_{k-1} - L_0\right], \quad (6)$$

$$L_K = \mathrm{D_{KL}}\left[q\left(\mathbf{u}_K \mid \mathbf{u}_0\right) \| p\left(\mathbf{u}_K|z\right)\right] \quad (7)$$

$$L_{k-1} = \mathrm{D_{KL}}\left[q\left(\mathbf{u}_{k-1} \mid \mathbf{u}_0\right) \| p_\theta\left(\mathbf{u}_{k-1} \mid \mathbf{u}_k, z\right)\right] \quad (8)$$

$$L_0 = -\log p_\theta\left(\mathbf{u}_0 \mid \mathbf{u}_1, z\right) \quad (9)$$

Here, the various components of the process exhibit a factorization that is akin to, yet more straightforward than, that observed in diffusion probabilistic models. The terms $L_{k-1}$ denote the Kullback-Leibler divergences between sequentially adjacent distributions.

$$\mathrm{D_{KL}}\left[q\left(\mathbf{u}_{k-1} \mid \mathbf{u}_0\right) \| p_\theta\left(\mathbf{u}_{k-1} \mid \mathbf{u}_k, \mathbf{z}\right)\right]$$
$$\propto \left\|\mu_\theta\left(\mathbf{u}_k, k\right) - \mathbf{H}\left(t_{k-1}\right)\mathbf{u}_0\right\|_2^2 \quad (10)$$

$$-\log p_\theta\left(\mathbf{u}_0 \mid \mathbf{u}_1, \mathbf{z}\right) \propto \left\|\mu_\theta\left(\mathbf{u}_1, 1\right) - \mathbf{u}_0\right\|_2^2 \quad (11)$$

Eq.(10) (11) $p_\theta\left(\mathbf{u}_{k-1} \mid \mathbf{u}_k, \mathbf{z}\right)$ $(k = 1, \ldots, T)$ are trainable Gaussians as Eq.(5). The losses on all levels are direct point reconstruction losses where we predict a slightly less smoothed points from a smoothed points that has added noise with $\sigma^2$.

## 3.2 Geometric Algebra Representation for Point Cloud

Present deep learning methodologies often handle the components of vector fields identically to scalar fields, aggregating all scalar fields along the channel dimension. This practice neglects the intricate geometric interplay between the various components within vector fields and between individual vector and scalar fields.

### 3.2.1 Background for Geometric Algebra Representation.

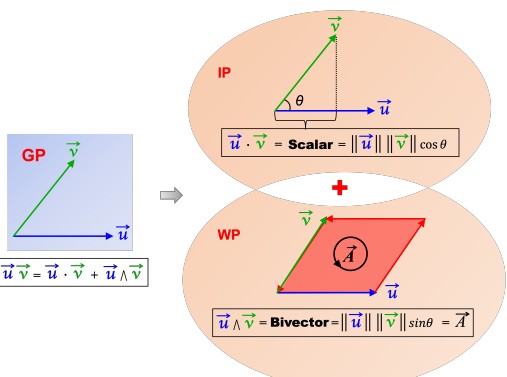

**Figure 2: The geometric product (GP) is composed of both the inner product (IP) and the wedge product (WP). The GP of two vectors yields a zero-dimensional scalar (length) and a two-dimensional bivector (oriented plane Ā).**

Geometric Algebra $G_{p,q}(\mathbb{R})$ is generated by $p + q$ orthonormal basis elements $e_1, \ldots, e_{p+q}$ generating vector space $\mathbb{R}^n$, such that the following quadratic relations hold as $e_i^2 = +1$ for $1 \leq i \leq p$, $e_j^2 = -1$ for $p < j \leq p+q$ and $e_i e_j = -e_j e_i$ for $i \neq j$. The pair $(p, q)$ is called the signature. By multiplying vectors, one obtains so-called multivectors, which can represent both geometrical objects and operators containing scalars, vectors, bivectors, $\cdots$, k-vectors. For example, in a $G_{3,0}(\mathbb{R})$ with orthogonal basis $e_1, e_2, e_3$, has $B(2^3 = 8)$ basis blades, a general multivector takes the form:

$$x = x_s + x_1 e_1 + x_2 e_2 + x_3 e_3 + x_{12} e_1 e_2 + x_{13} e_1 e_3$$
$$+ x_{23} e_2 e_3 + x_{123} e_1 e_2 e_3, \quad (12)$$

These are characterized by their dimensionality (grade k), such as scalars (grade 0), vectors $e_1$(grade 1), bivectors $e_1 e_2$(grade 2), tirvectors $e_1 e_2 e_3$(grade 3).

In geometric algebra, multiplication is achieved through the geometric product, which encompasses both the inner product and the wedge product. As a fundamental component of geometric algebra, the geometric product merges the dimension-reducing properties of the inner product—which converts a vector into a scalar, representing length—with the dimension-expanding attributes of the outer product. Specifically, the outer product of two vectors produces a bivector, a directed area equivalent to the area of the parallelogram formed by these vectors, oriented perpendicular to their plane. Consequently, the geometric product retains angular relationships between vectors and captures normal information of the plane they span, offering a more comprehensive representation than the inner product alone, as depicted in Figure 2.

### 3.2.2 Geometric Algebra Representation for point cloud.

For this task, we aim to utilize a novel point transformer-based architecture[22, 45], incorporating $G_{3,0}(\mathbb{R})$ Representation for feature expression. Consequently, modifications are necessary for point cloud embedding, the linear layers within the transformer, and the attention mechanisms to accommodate this approach.

We first embedding the point cloud position lies in vector space $X^{N \times 3} \in \mathbb{R}^3$ into $X^{N \times C \times B} \in G_{3,0}(\mathbb{R})$. This step is quite straightforward; it simply involves embedding the elements of a vector space into a multivector space according to their corresponding positions.

We further define the geo-linear layers to perform feature mapping act as the linear layers in classical transformers, in which multivector features are mapped into $X^{N \times C_0 \times B} \in G_{3,0}(\mathbb{R})$. The geo-linear is formualted as:

$$X_{c_1}^{(k)} = \sum_{c_{in}=1}^{\ell} W_{c_{out}c_{in}k}^1 X_{c_{in}}^{(k)},$$

$$X_{c_2}^{(k)} = \sum_{c_{in}=1}^{\ell} W_{c_{out}c_{in}k}^2 X_{c_{in}}^{(k)}, \quad (13)$$

$$X_{out}^{(k)} = \sum_{i=0}^{n} \sum_{j=0}^{n} W_{ijk} \left\{ x_{c_1}^{(i)} x_{c_2}^{(j)} \right\}^{(k)},$$

where the first two lines represent feature mapping along the grade levels within the multivector, the last line represents the element-wise geometric product layer. $W_{c_{out}c_{in}k}$ are learnable coefficients. Their geometric product terms take the form $\left\{ x_{c1}^{(i)} x_{c2}^{(j)} \right\}^{(k)}$.

With the incorporation of this final element, we establish the query, key, and value multivector representations $X^{N \times C \times B}$, thus finalizing the attention mechanism for multivectors as:

$$\text{Attention}(q, k, v)_{i'c'} = \sum_{i} \text{Softmax}_i \left( \frac{\sum_c \langle q_{i'c}^{MV}, k_{ic}^{MV} \rangle}{\sqrt{n}} \right) v_{ic'} \quad (14)$$

Where $\langle \cdot, \cdot \rangle$ denotes the inner product in geometric algebra, which facilitates efficient computations. Additionally, we have extended this attention mechanism to accommodate a shared multi-head self-attention structure, utilizing a method akin to the one outlined in [3]. Since the network ultimately outputs a vector, all that is required is to perform the inverse of the embedding operation.

## 3.3 Training and Inference

### 3.3.1 Training.

As mentioned earlier(Section 3.1.2), the model optimize the reconstruction losses on all time levels directly. In inverse heat dissipation process, the ability to easily obtain a one-to-one mapping between point clouds across adjacent time segments allows the model to directly optimize the reconstruction losses at all temporal levels. Consequently, the Mean Squared Error (MSE) loss is employed due to its simplicity and effectiveness in mapping relationships. Furthermore, the Earth Mover's Distance (EMD) loss has been demonstrated to exhibit robust and exceptional reconstruction performance in point cloud processing. Therefore, the final loss function $f$ is formulated by combining these two losses:

$$f(s, t) = MSE(s, t) + \lambda EMD(s, t) \quad (15)$$

$$Loss = f(\boldsymbol{\mu}_\theta(\mathbf{u}_k, k, \mathbf{z}), \mathbf{H}(t_{k-1})\mathbf{u}_0) \quad (16)$$

where $\lambda$ means a weighting factor ( $\lambda = 1$ in this paper).

### 3.3.2 Inference.

We employ the completed forward process of interpolated points as the initial distribution $p_\theta(\mathbf{u}_K \mid \mathbf{z})$, feeding it into the network, which significantly enhances the quality of upsampling during inference:

$$\mathbf{u}_{k-1} \leftarrow \boldsymbol{\mu}_\theta(\mathbf{u}_k, k, \mathbf{z}) + \delta^2 \boldsymbol{\varepsilon}_k \quad (17)$$

After $k$ iterations, we obtain the mean of the final distribution, $p_\theta(\mathbf{u}_0|\mathbf{z})$ (with the noise bias removed in the last step), which represents the result of the final point cloud upsampling.

## 4 EXPERIMENTS

### 4.1 Experiment Setup

**Dataset.**

In our experiment, we employ two well-established public benchmarks for evaluation purposes: PUGAN[14] and PU1K[26]. We follow the official training and testing partition protocols provided for these datasets. Utilizing Poisson disk sampling[44], we generate training datasets comprising 24,000 and 69,000 uniformly distributed patches, respectively. Each training patch is composed of 256 points, and its corresponding ground truth contains $256 \times R$ points. For the testing phase, all testing low-res point clouds from both datasets have 256 points, while the high-res counterparts contain $256 \times R$ points.

In addition to the aforementioned synthetic datasets, we also utilized mesh data [37] and real-scanned 3D Mobile Laser Scanning (MLS) data [34] for both quantitative and qualitative evaluation.

**Baselines and Evaluation Metrics.**

we have trained PU-Net[43], PU-GAN[14], PU-GCN[26], Meta-PU[41], Grad-PU [6] [29] and Neural Points (NePs)[4] with the default settings in the respective papers as baselines.

Following pioneers, we adopt the Chamfer distance (CD), Earth mover distance (EMD) and Hausdorff Distance (HD) as metrics. For all the metrics, the smaller the metric, the better the quality of the results.

## 4.2 Results and Comparisons

We integrated the test results of PU1K[26]and PUGAN[14]into a single dataset to save space and provide clear insights. The results and comparisons of 4× and 16× are given in Table 1, Figure 3 and Figure 4. Our method achieves the best performance both

| model | PU1K | | | | | | PUGAN | | | | | |
|---|---|---|---|---|---|---|---|---|---|---|---|---|
| up ratio | 4x | | | 16x | | | 4x | | | 16x | | |
| model | CD↓ | HD↓ | EMD↓ | CD↓ | HD↓ | EMD↓ | CD↓ | HD↓ | EMD↓ | CD↓ | HD↓ | EMD↓ |
| PU-NET | 7.649 | 0.371 | 11.97 | 14.07 | 1.853 | 22.11 | 35.80 | 1.538 | 55.48 | 66.43 | 8.30 | 102.8 |
| PU-GAN | 7.894 | 0.5788 | 13.57 | 5.261 | 0.7407 | 10.58 | 32.79 | 4.074 | 56.38 | 21.75 | 3.793 | 42.96 |
| PU-GCN | 6.661 | 0.5054 | 19.10 | 6.048 | 0.6019 | 12.26 | 29.92 | 1.967 | 4.615 | 27.01 | 2.450 | 55.04 |
| Meta-PU | 6.237 | 0.4199 | 10.33 | 3.349 | 0.4172 | 8.008 | 25.91 | 1.346 | 4.607 | 14.28 | 1.917 | 33.56 |
| Grad-PU | 13.48 | 1.486 | 21.917 | 8,922 | 1.276 | 23.57 | 50.49 | 5.703 | 12.68 | 34.96 | 4.468 | 89.95 |
| NePs | 5.195 | 0.445 | 9.232 | 3.287 | 0.5630 | 7.249 | **2.332** | 1.304 | 4.060 | 13.81 | 1.645 | 29.29 |
| Ours | **4.877** | **0.4138** | **7.566** | **2.316** | **0.438** | **5.056** | 2.347 | **1.198** | **3.951** | **9.505** | **1.555** | **18.69** |

Table 1: Results and comparisons for $4\times$ and $16\times$ upsampling, which metrics CD($\times10^{-6}$), HD($\times10^{-4}$), EMD($\times10^{-6}$).

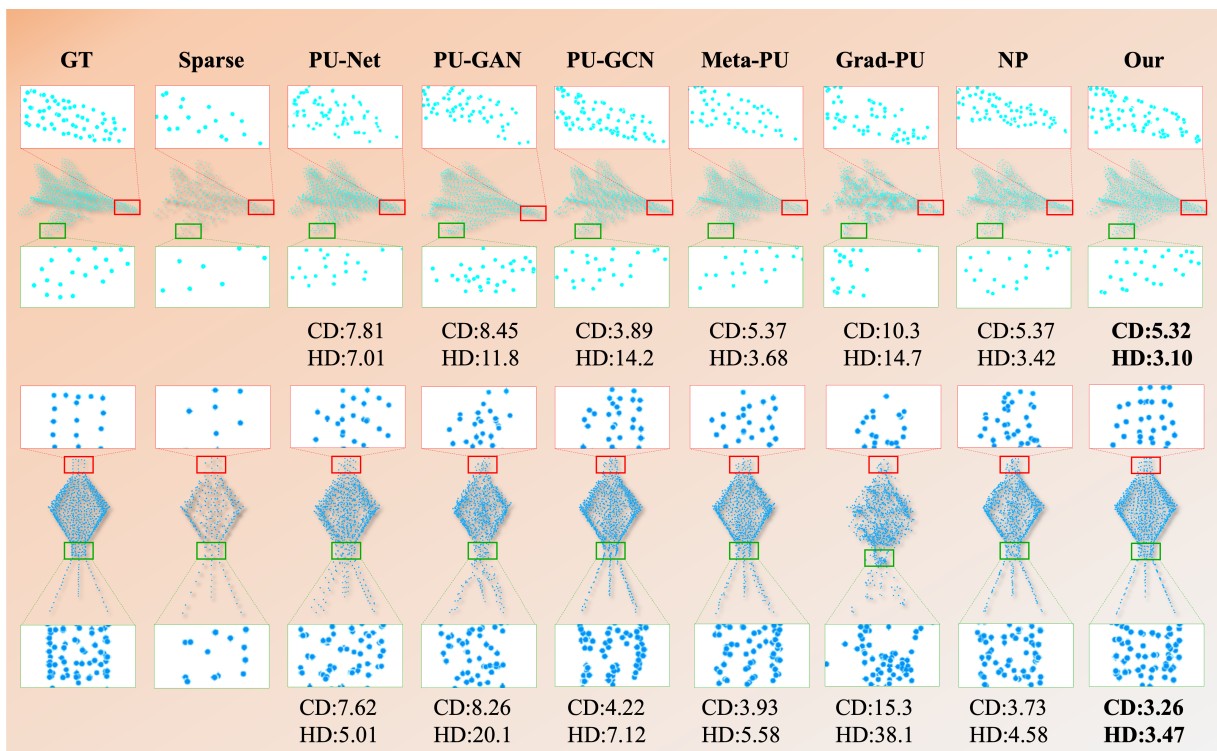

Figure 3: $4\times$ results and comparison on the PU1K[26]dataset, with error metrics including CD($\times10^{-6}$) and HD($\times10^{-3}$). Some chosen viewpoints are selected to highlight the detailed improvements. The same visualization approaches apply to subsequent sections.

quantitatively and qualitatively. The outcomes from PU-Net[43] are generally disorganized. While PU-GAN[14] outperforms PU-Net[43], it still introduces some unusual noise and outliers. PU-GCN[26] surpasses both in performance, effectively maintaining flat areas while only generating minor noise points within regions abundant in features. We exerted our effort to train Grad-PU [6] using the default parameters; however, the outcomes fell short of expectations, nearly resulting in the loss of a significant portion of the point cloud's geometric information. Meta-PU[41] and Neps[4] are recently point cloud upsampling methods with outstanding performance, and the authors have generously provided the pre-trained model parameters. The point cloud distributions generated by Meta-PU[41] are consistently uniform, often surpassing our expectations to some degree. However, in regions of high curvature,

Meta-PU struggles to produce high-quality results. Neps[4] achieves performance closest to ours in terms of results, yet it falls short in handling details perfectly. Additionally, the Neps[4] network incorporates normal vector information of point clouds during training. Thus, our significant advantage lies in accomplishing point cloud upsampling without relying on normal vectors.

## 4.3 Ablation Study

We conduct ablation studies to demonstrate the effectiveness of our proposed method and to illustrate how each component contributes to the final results.

Firstly, we aimed to validate the diffusion-type network paradigm. Our focus was on determining if enhancement based on reverse heat dissipation geometry resolution surpasses traditional

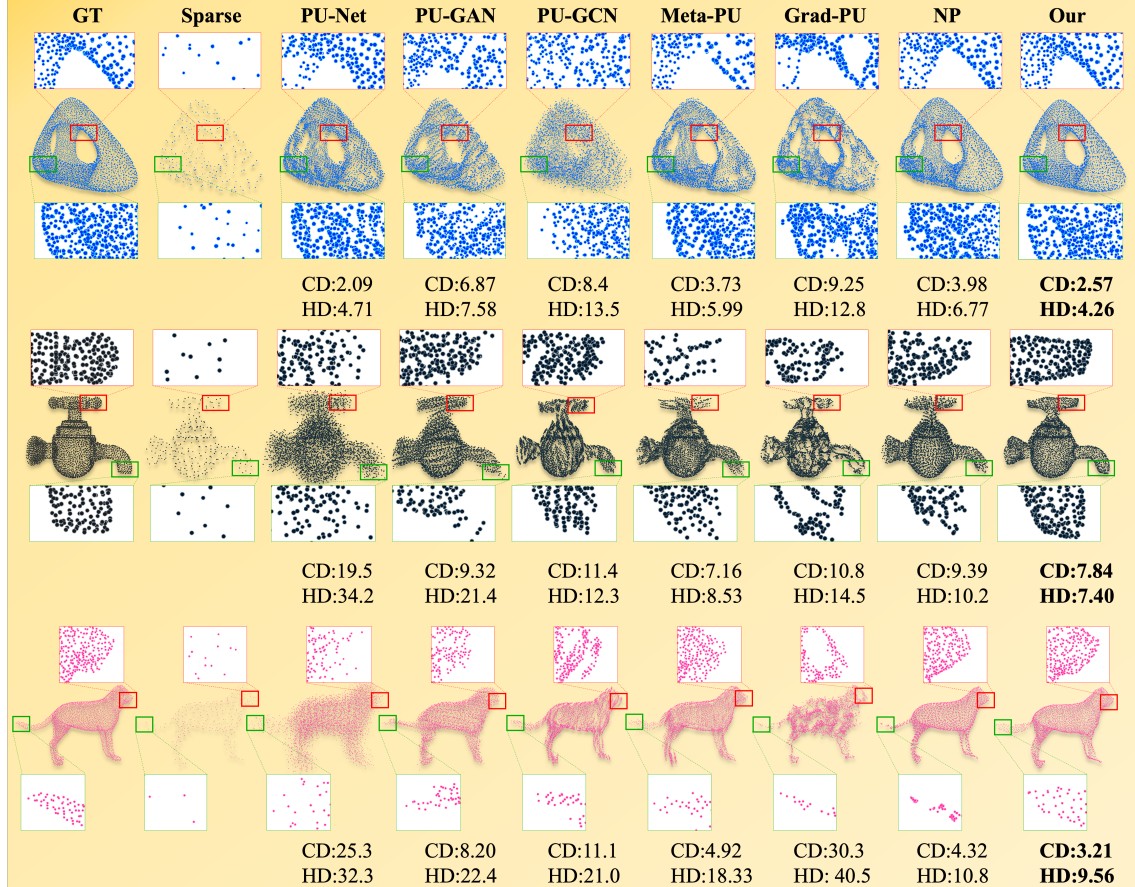

**Figure 4:** 4× **upsampling results and comparison on the PUGAN[14]dataset, with error metrics including CD**($\times 10^{-6}$) **and HD**($\times 10^{-3}$)**.**

denoising approaches. This concept bears similarities to the ideas presented in [29]. However, due to the absence of source code from the authors, we defined the hyperparameters and fine-tuned a Conditional DDPM for Point Cloud Upsampling ourselves. Secondly, we aim to validate whether our proposed approach of extending features into the multivector space through a geometric algebra representation(GAR) offers a geometric enhancement advantage over the traditional method of features lying in vector spaces(VSF). We can see that the result with the conditional DDPM scheme is much worse than the inverse dissitation scheme. In the comparison between Geometric Algebra Representation (GAR) and Vector Space Features, GAR demonstrates superior advantages, both quantitatively and qualitatively. The evidence supporting these arguments is verified in Table 2. It is particularly noteworthy that within a multivector, elements of different grades can participate in geometric multiplication.

### 4.3.1 Generalization and Robustness.
**Generalizing Across Different Point Cloud Sources.**

We tested the 4× and 16× upsampling performance on the PU1K and PUGAN test sets, which utilize synthetic data, and we used

| Ablation Settings | CD↓ | HD↓ | EMD↓ |
|---|---|---|---|
| Conditional DDPM + VSF | 6.08 | 9.88 | 14.1 |
| Inverse Heat + VSF | 4.32 | 4.37 | 9.85 |
| Inverse Heat + GAR | **3.42** | **2.31** | **8.99** |

**Table 2: Results of the ablation study on mesh data [37], with** 4×**upsampling factor.which metrics CD**($\times 10^{-6}$)**, HD**($\times 10^{-3}$)**, EMD**($\times 10^{-6}$)

mesh data in the previous section's ablation study. We also conducted experiments on real-scanned point clouds from the Paris-rue-Madame database[34]. Given that these point clouds contain millions of points, we limited our comparison to qualitative assessments and excluded methods that do not support inference on such large-scale point clouds or consistently yield inferior results. Scanned data are frequently sparse and noisy, and often contain small holes or gaps that compound their complexity. Figure 5 demonstrates that our results are more complete, smooth, and accurate, whereas other methods often retain these holes.

**Robustness to different upsampling factor.**

We examine the upsampling effects of various techniques using large upsampling factors. It should be noted that all methods tested,

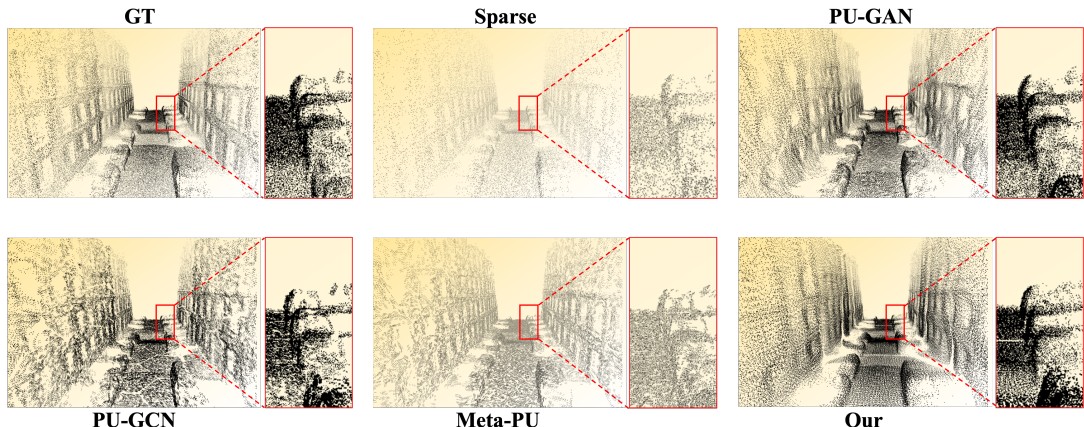

**Figure 5: Upsampled results on the Paris-Rue-Madame dataset. This dataset features radar scenes of streets and roadside parking. Our upsampling method closely approximates the ground truth.**

including ours, are trained with a 4× supervision data. Due to the simplicity of Midpoint Interpolation, we can easily generate a coarse upscaled point cloud, facilitating the training of an Rx supervision signal if conditions permit. In fact, due to the high computational costs of 4× supervision training, we use the model twice to achieve 16× upsampling, still yielding favorable outcomes. However, the cascading upsampling approach employed by PU-GAN, PU-GCN, and Grad-PU exhibits noticeable flaws. Therefore, if we train a model under R× supervision, in theory, we could achieve point cloud upsampling at any integer power of R.

**Robustness to different point density.**

We continue to investigate the upsampling effects of different techniques with different point density. In this context, density primarily refers to the number of points per unit space within the point cloud that is to be upsampled, compared to the 256-point clouds used as input during training. Excluding the Neps model, which cannot alter input sizes, and the PU-Net, which shows a significant performance gap, our model remains highly competitive across various input densities as described in Table 3.

| | 512 points | | | 1024 points | | | 2048 points | | |
|---|---|---|---|---|---|---|---|---|---|
| Model | CD↓ | HD↓ | EMD↓ | CD↓ | HD↓ | EMD↓ | CD↓ | HD↓ | EMD↓ |
| PU-GAN | 15.2 | 1.47 | 26.4 | 4.06 | 0.61 | 7.16 | 2.28 | 4.49 | 4.18 |
| PU-GCN | 7.21 | 0.66 | 12.6 | 4.38 | 0.45 | 7.80 | 2.91 | 0.32 | 5.29 |
| Meta-PU | 5.57 | **0.54** | 10.8 | 2.81 | **0.33** | 5.63 | 1.38 | **0.15** | 2.78 |
| Grad-PU | 5.91 | 0.72 | 12.7 | 2.89 | 0.44 | 5.75 | 1.41 | 0.20 | 2.77 |
| Ours | **5.54** | 0.75 | **10.2** | **2.72** | 0.42 | 5.40 | **1.20** | 0.23 | **2.73** |

**Table 3: 4× upsampling results and comparisons for 512, 1024 and 2048 inputs, which metrics CD($\times 10^{-6}$), HD($\times 10^{-4}$), EMD($\times 10^{-6}$).**

## 4.4 Conclusion and Limitation

In summary, this paper introduces a novel point cloud upsampling method based on PDEs, tailored specifically for dense point clouds through a forward heat dissipation process and its refined inverse. By integrating geometric algebra to articulate complex geometric features within multivector fields, our approach not only significantly enhances the precision and expression of geometric characteristics but also surpasses existing techniques in generating

intricate geometric forms. Extensive experimental validations underscore the superiority of our method. Moreover, the successful application of inverse heat dissipation and geometric algebra in this context not only proves effective for point cloud upsampling but also paves the way for future advancements in handling complex geometric data, potentially benefiting a wide range of applications from 3D modeling to autonomous navigation.

Currently, our diffusion-type point cloud upsampling method requires a longer inference time to achieve enhanced upsampling results. This is despite the fact that our supervised training approach significantly reduces the number of inference iterations compared to DDPM's denoising scheme. Moreover, the use of geometric algebra representation through geometric products also demands substantial computational resources. Therefore, in upsampling tasks with particularly high upsampling factors, it is necessary to reconsider the balance between performance and computational efficiency.

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
