# OpenReview forum: "Point Cloud Upsampling With Geometric Algebra Driven Inverse Heat Dissipation"
_acmmm.org/ACMMM/2024/Conference — MM2024 Oral_

### Official Review · Reviewer_zhau · 2024-05-23

**Rating:** 6
**Confidence:** 3

**Summary:**

Based on the intuition of erasing point clouds to maintain shapes, the author proposes a novel and effective point cloud upsampling method combining PDE and multi-vector fields. Sufficient experimental results show that the method's modeling capabilities in point cloud geometric details significantly surpass existing methods.

**Strengths:**

1. Well-organized paper with clear structure and smooth logic.
2. The author proposes a PDE-based heat dissipation forward process, which can better retain the geometric details of point clouds during the diffusion process and generate high-quality point clouds.
3. The author proposes to use multi-vector fields to represent point clouds, which further enhances the modeling ability of point geometric characteristics.
4. The proposed method performs well in a complete experimental setup. The experiment is complete and the discussion is sufficient. The author explains the experimental details in detail.

**Limitations:**

1. It is recommended to add a detailed flow chart of the method to improve the reproducibility of the work. Without it, the full picture of the algorithm is difficult to imagine, and it is recommended to show the flow of features from input to output.
2. Comparison with the DDPM-based method and the transformer-based method can better reflect the contribution and advantages of the paper's work. However, the experiments in Section 4.2 do not fully reflect the comparison with these methods. For this reason, it is recommended to add visualization results of the ablation experiment.

**Suitability:**

3

---

### Official Review · Reviewer_fit3 · 2024-05-23

**Rating:** 5
**Confidence:** 3

**Summary:**

This work leverages a new diffusion scheme, the inverse heat dissipation model for the point cloud upsampling task. For more effective point feature expressiveness, it further introduces a geometric algebra representation to expand the feature space into multivector fields.

**Strengths:**

1. Introducing the PDE-based heat dissipation into the 3D point cloud domain.
2. The idea of expanding the point feature space has its novelty and the visualized results seem good.
3. The writing is clear and well-structured. The reasoning about why replacing DDPMs with the PDE-based heat dissipation model is convincing and the writing of the Method part is solid.

**Limitations:**

1. The experiment part provides qualitative and quantitative comparisons with classical methods like PU-Net and PU-GAN.  However, including more comparison results with the latest methods is strongly recommended.

2. Some typos in the writing and tables, e.g., typos in line 295 and line 308, missing bold identifier in column 7, Table 3.

3. How exactly does the model realize arbitrary upsampling at flexible scales is not mentioned.

**Suitability:**

3

---

### Official Review · Reviewer_5KDo · 2024-05-24

**Rating:** 4
**Confidence:** 2

**Summary:**

This paper proposes a novel method for upsampling point cloud, driven by geometric algebra and the inverse heat dissipation process. By integrating geometric algebra with Partial Differential Equations (PDEs), the method simulates the inverse heat dissipation process within point clouds, replacing traditional noise addition techniques. The proposed approach was evaluated on the PUGAN and PU1K public datasets, demonstrating SOTA performance in 4x and 16x upsampling tasks.

**Strengths:**

1. The paper introduces a novel approach by combining geometric algebra with the PDE model of the inverse heat dissipation process for the task of upsampling point clouds.
2. By utilizing multivectors as the representationfor point cloud features, the method effectively captures and expresses the complex relationships between different dimensions within the point cloud, which enhances the modeling capability for complex geometric structures.
3. The proposed method demonstrates impressive performance across multiple datasets.

**Limitations:**

1. While the use of diffusion for point cloud recovery is relatively underexplored, the technique itself is not novel. Approaches from image inpainting or superresolution can likely be adapted to this task and should be considered for comparison.

**Suitability:**

2

---

### Official Review · Reviewer_u3ov · 2024-05-24

**Rating:** 4
**Confidence:** 2

**Summary:**

This paper proposes a new method for point clouds upsampling. In order to better capture the fine-grained geometric details of the point clouds, the proposed method upsamples the point cloud by inverting the heat equation, a PDE that locally erases fine-scale details [1]. In addition, the method uses multivectors from geometric algebra to enrich the representations of the point cloud. Experimental results show that the proposed method outperforms the compared methods and its design choices are validated.

**Strengths:**

* The idea of modeling point cloud upsampling as inverse heat dissipation seems to be new. Also, representing point clouds with multivectors from geometric algebra is new to point cloud upsampling.
* The proposed method has achieved competitive performance against the compared methods.
* Two main designs of the proposed method, inverse heat dissipation and multivector representation, are proved to be effective.

**Limitations:**

Comments and Questions
* There seems to be a lot of related work on learning based point cloud upsampling (Section 2.2) missing. Please see the reference list below ([2] - [16]) for some examples.
* More existing methods should be added to Table 1, considering that there has been a lot of work on point cloud upsampling recently. Can the authors additionally compare the proposed methods with some recent works like [15] and [16] in the list below?
* Can the authors compare the inference speed and model size of the proposed with other works?

Minor comments
* In Table 1, if bold text describes the best performance in a row, then 0.4172 in the row "PU1k-16x-HD" should be bold and 0.438 should be underlined.

[1] GENERATIVE MODELLING WITH INVERSE HEAT DISSIPATION

[2] Point Cloud Upsampling via Disentangled Refinement

[3] PU-Transformer: Point Cloud Upsampling Transformer

[4] Semantic Point Cloud Upsampling

[5] PU-EVA: An Edge-Vector based Approximation Solution  for Flexible-scale Point Cloud Upsampling

[6] SSPU-Net: Self-Supervised Point Cloud Upsampling via Differentiable Rendering

[7]PC2 -PU: Patch Correlation and Point Correlation for Effective Point Cloud Upsampling

[8] PUFA-GAN: A Frequency-Aware Generative Adversarial Network for 3D Point Cloud Upsampling

[9] Parametric Surface Constrained Upsampler Network for Point Cloud

[10] iPUNet: Iterative Cross Field Guided Point Cloud Upsampling

[11] Deep Magnification-Flexible Upsampling over 3D Point Clouds

[12] APUNet: Attention-guided upsampling network for sparse and non-uniform point cloud

[13] PU-GAT: Point cloud upsampling with graph attention network

[14] SSPU-Net: A Structure Sensitive Point Cloud Upsampling Network with Multi-Scale Spatial Refinement

[15] A Noising-Denoising Framework for Point Cloud Upsampling via Normalizing Flows

[16] Self-Supervised Arbitrary-Scale Implicit Point Clouds Upsampling

**Suitability:**

2

---

### Meta-Review · Area_Chair_wVua · 2024-07-01

**Recommendation:** Accept (Oral)
**Confidence:** 5

**Metareview:**

This paper proposes a method for point cloud upsampling via geometric algebra and the inverse heat dissipation process. All reviewers appreciate the idea and experimental results. Therefore, the paper is recommended to be published in ACM MM. However, Reviewer 5KDo has concerns to the missing important references including "Generative Modeling With Inverse Heat Dissipation" and "Blurring Diffusion Models". The authors should cite and add the discussion with these related works in the camera ready version.